# Mapping Accessibility in the Historic Urban Center of Bucharest for Earthquake Hazard Response

Ioan Ianoș<sup>1</sup>, George-Laurentiu Merciu<sup>2</sup>, Cristina Merciu<sup>1</sup>, and George Pomeroy<sup>3</sup>

<sup>1</sup>Interdisciplinary Center of Advanced Research on Territorial Dynamics, University of Bucharest, Blvd. Regina Elisabeta, 4-12, code 030018, Romania

<sup>2</sup>Faculty of Geography, University of Bucharest, Blvd. Nicolae Bălcescu, 1, code 030018, Romania
 <sup>3</sup>Geography – Earth Science Department, Shippensburg University of Pennsylvania. 1871 Old Main Drive, Shippensburg PA 17257

Correspondence to: George Pomeroy (gmpome@ship.edu).

Abstract. Planning for post-disaster accessibility is essential in providing emergency and other services to protect life and property in the impacted areas in the aftermath of such events. Careful planning is especially critical in congested historic districts where narrow streets and at-risk structures are more common or prevail. One of the more common methods of measuring accessibility, the use of isochrones, may be particularly inappropriate in congested historic areas. Bucharest,

- Romania, is a city with such an older historic core of buildings and narrower streets. Furthermore, Bucharest ranks behind only Istanbul among large European cities in terms of seismic risk. The city experienced earthquakes of magnitude greater than 7.0 in 1940 and 1970. Both earthquakes had their epicenter in the Vrancea Mountains of central Romania and less than 200 kilometers from Bucharest. With a relative periodicity of 45 to 50 years for earthquakes of such magnitude, there is a clear need for earthquake hazard planning that incorporates comprehensive hazard awareness, including an improved understanding
- of and mapping for accessibility, particularly in areas with greater potential for building collapse. This paper provides an examination of simulated accessibility for central Bucharest through the use of mapping and GIS technologies. The finding is that accessibility will be substantially compromised by anticipated building collapse. Therefore, policy makers and planners need to more fully understand and appreciate the serious implications of this compromised accessibility in planning for emergency services and disaster recovery activities.

# 25 1 Introduction

A longitudinal analysis of natural hazards that impact cities shows an increasing awareness of the frequency of disasters and especially earthquakes (Eshghi & Larson, 2008; Armaş, 2012; Lu & Xu, 2014). Their effects demand a prompt response from decision makers and the population, through the proper management of the emergency situations (Waugh & Streib, 2006). Earthquakes are among the natural disasters that generate the greatest human and material losses (Geis, 2000; Armaş & Avram,

2008; Atanasiu & Toma, 2012).

Many areas of high seismic risk are urbanized and densely populated (Pollino et al., 2012; Vatseva et al., 2013). In addition and coincidentally, many countries in transition are characterized by urban growth is uncontrolled, including a number of large

25

and medium-sized urban centres where growth is especially chaotic (Salvati, 2014). Thus, an increase in such disasters and their impacts can reasonably be anticipated. Compounding this is the fact that man new buildings, new structures, and sometimes newer infrastructure elements, under developers' pressure, are frequently out of compliance with the construction regulations established for different seismic areas. In addition, the large time lag between two strong earthquakes (Schweier

& Markus, 2009) dulls public awareness as to the potential impacts of such disasters. Also, those in charge of the emergency situations management sometimes become complacent.

The flow and urban structures analysis shows that vulnerability to earthquakes is one that requires a separate approach from other types of disasters (Armaş, 2008; Bostenaru Dan & Armaş, 2015). This is because, unlike disasters that can be anticipated in the short term (such as storms) there is little or no delay between the occurrence of the earthquake and the subsequent loss

- of life and property damage that occurs. Therefore, emergency response activities must be executed very quickly and efficiently (Wegscheider et al., 2013). For cities with an earthquake risk, an important ally is the public perception over such events which, once educated towards a quick response, can help reduce property damage and especially the number of casualties (Armaş & Avram, 2008). Meanwhile, it is known that, no matter how well organized the mitigation process of the disastrous effects of earthquakes can be, still they cannot be avoided (Momani & Salmi, 2012).
- In recent years, seismic risk management has been more fully studied and developed so as to establish a series of priorities related to the rehabilitation of buildings considered to be of major importance, including schools (Crowley et al., 2008; Raffaelle et al., 2013; Panahi et al., 2013), public institutions, historical buildings, and monuments (Grasso & Maugeri, 2009; Pessina & Meroni, 2009). There is a demonstrated need for coherent urban policy (Ianoş et al., 2017) which would mitigate the occurrence of blockage points during emergency interventions.
- In emergency situations, the key element is rapid accessibility to places where possible casualties may be located. Timely intervention within the first two hours is critical in saving the wounded and in determining the safest access routes for specific equipment proves essential.

In general, natural disaster management includes actions that are particularly important for communities vulnerable to such events, which lead to the development of impact scenarios before the natural hazards occur (Bakillah et al., 2013). In this sense, GIS techniques may be particularly useful in developing decision-making scenarios for potential earthquake disasters.

Our study shows that a special attention should be paid to accessibility in the historic centers of large cities (Ianos & Cepoiu, 2009). Starting from the idea that the historic centers of cities are characterized by both intense pedestrian traffic and by attractive older buildings that are by their historic nature also very dangerous, we can reasonably speculate that determining the accessibility in terms of emergency situations will facilitate quick intervention in areas with injured people, casualties or

30 earthquake-related phenomena (fires, gas accumulations, local flooding). The main objective of the study is to integrate the geospatial data in thematic mapping products and to use GIS techniques in order to provide solutions for seismic risk management within Bucharest.

Unlike most studies that relate to communities' response after an earthquake occurrence, the critical analysis of the emergency situations management generated by it (Pollino et al., 2012; Wegscheider et al., 2013; Lu & Xu, 2014), the present

study demonstrates the importance of GIS analyses in detecting potential congestion and inaccessibility issues in areas where buildings will likely collapse as a result of an earthquake.

## 2. Case Study

- Bucharest is Romania's largest city (1.9 million inhabitants), national capital, and one of the great metropoles of the southeastern Europe (Grosse-Espon project, 2014). Its urban evolution was very rapid, especially starting with the second half 5 of the 19th century. Currently, the city occupies an area of 228 square kilometers, possessing a housing stock predominantly consisting of multifamily buildings of "block" type, built during the communist period (Ianos et al., 2015). Located about 135 km from the epicenter of Vrancea seismic area (Lungu et al., 2000), in close proximity to the Southern Carpathians and along the contact between Eastern European, intra-Alpine and Moesian plates (Mărmureanu et al., 2011), the city is extremely vulnerable to earthquakes. Indeed, in a classification of the European metropolitan areas with respect to potential loss of life 10
  - and damage to property, Bucharest is ranked second after Istanbul (Bala, 2013).

The historical record of Bucharest is replete with accounts of damaged earthquakes since the city's founding (Tatevossian & Albini, 2010), having the epicenter in Vrancea area. The Vrancea seismic area is responsible for the highest seismic risk in Romania (Ardeleanu et al., 2005). Over the past 76 years, Bucharest was affected by four earthquakes with a magnitude

15 between 6.9 and 7.7 on the Richter scale (November 1940, March 1977, August 1986 and May 1990). It is interesting that the decline in seismic waves is much stronger to the north and west than towards south, to Bucharest (Pavel et al., 2014).

The study area for this paper is confined to central Bucharest, an area of approximately 8.33 square kilometers (Figure 1). The oldest part of the city is situated in the south of this area, grouping together the historic center (the 16th-17th centuries); the central and northern parts date to the 18th-19th centuries. All the earthquakes that occurred during the interval mentioned

above have especially impacted these areas, the most powerful effects being caused by the earthquakes during 1940 and 1977. 20 The earthquakes that followed over the time caused a natural selection of the buildings, thus explaining the small number of buildings dating back more than 200 years.

The most serious problem of this area is the large stock of buildings from the late 19th century (within the historic center), which have structurally deteriorated over time and they no longer meet the current standards related to seismic risk. Not only

25 does Bucharest exhibit a high level of exposure to earthquake hazards, it also suffers from inefficiently organized civil protection services and a low level of public awareness and education concerning these seismic risks (Armas, 2006).

Experience has shown this area of Bucharest to be most vulnerable. Memories from the strongest earthquake which took place in 1977 bear an increasingly louder echo as the anticipation and anxiety build concerning the next big earthquake. In essence, there is a fear that the city will be no better prepared than it was in 1977 (Figure 2) and there is a recognition that greater preparedness is needed.

30

Immediately after the earthquake of 4 March 1977, the former regime announced the start of a rehabilitation project of the highly-degraded buildings within the central area, a project which was abandoned in less than a year. Many buildings, after being braced in position for 6-7 months with wooden or metal poles which were later withdrawn, were then only "cosmeticized"

5

and then reoccupied. In making such decisions, a precedent for irresponsible policy was set that, unless addressed and altered, will have disastrous long-term consequences. Additionally, time has been a great enemy and a permanent state of vigilance is needed. Finally, there is also a need for earthquake mitigation related adequate public investments in the areas most vulnerable to earthquakes. Until there is wider public acknowledgment of these high seismic risk buildings (as a result of surveys conducted in the mid-90s and even subsequently) the apartments in these blocks were still the most expensive due to their central location and their spaciousness.

#### 3. Data and Methods

The assessment of the seismic hazard and vulnerability includes quantitative and qualitative data analysis that incorporates physical, environmental, social and economic factors, as well as consequences, given that risk maps are known and the population to be potentially affected may also be estimated (Mandrescu, 1990; Rufat, 2013; Pascale, 2012).

### 3.1 Data

The authors have used several data sets (buildings subject to seismic risk, the presence of hospitals, and the presence of fire stations) in order to highlight the realistic dimension of the impact that a potential earthquake could have in the historical center of Bucharest municipality. Only those fire stations and hospitals in the municipal limits of Bucharest are considered (Table 1).

To represent the accessibility patterns prior and subsequent to the earthquake, it was necessary to digitize all elements of transport infrastructure, constructions, green spaces, alleys, sidewalks, property limits.

- Several different map sources were used to identify building locations including cadastral maps at scales of 1: 500, IGFCOT, 1: 2000 IGFCOT (1974-1975). Other map types and sources include old maps of Bucharest offered by the Topographic Military Direction, orthophotomaps (2014) taken from the National Agency for Cadastre and Real Estate Advertising. Also, the authors have overlapped the accessibility patterns on the numerical model of the land, resulting the absence of the barrier from the point of view of the relief, due to the fact that the Bucharest municipality is located in an area of plains.

## 3.2 Methods

Mapping the accessibility of the central area of Bucharest, an area highly vulnerable to earthquakes, was completed using GIS techniques incorporating spatial analysis. The calculation of accessibility was based on the geometric structure of the public

transport network (busses, trams and underground services), walking and cycling networks (Graeme & Aylward, 1999; Parker & Campbell, 1998; Naphtali, 2006; Svensson, 2010; Weiping & Chi, 2011; Sotoudehnia & Comber, 2011; ESPON TRACC Interim Report, 2013; ESPON GROSSE, 2013; Blandford, Kumar, Luo & MacEachren, 2012; Coffee et al., 2012; Yiannakoulias, Bland, Svenson, 2013; Vojnovic et al., 2014).

After modeling the road network using the ArcGIS Network Analyst extension, the authors used an assortment of analytical tool. These include use of the New Route tool was to check the road network; the New Closest option was able to determine the closest construction to each point (hospital, fire department); and with the New OD matrix function, the authors managed to determine optimal routes (depending on road distance and travel time) following the principle of the shortest possible route to establish links between each pair of points.

To highlight the accessibility in the most comprehensive way, the street structure (which is very dense in the historic center and where the streets are narrow) and road traffic must be taken into account. The accessibility was calculated as a function of the distances between different residential areas and hospitals areas and of the time necessary for these movements (using isochrones). Isochrone maps, showing travel times by public transport from the city centre, were used to assist urban transport

planning in the 1950s (Kok 1951, Rowe 1953 quoted by O'Sullivan, et al., 2000). These isochrones were generated using geographic information systems (GIS).

Accessibility was calculated to take into account the existence of specific service locations which could amplify the potential disasters, such as gas stations and electric transformers (Rezaie & Panahi, 2015). Also, the Kriging Kernel interpolation calculation and local polynomial interpolation were used. For exact interpolation, inverse the distance weighted (IDW) method

was used. In the scientific literature, "access" is mainly measured as a physical distance or travelling time (Sotoudehnia & Comber, 2011).

Thus, there have been identified support elements for more proactive management that will lead to a decrease in material damage and human casualties in case of strong earthquakes occurrence. Using a database associated to GIS environment represents a way to assess and estimate the potential damage that can be caused by such an event. At the same time, GIS is a

20 valuable method of analysis because the databases can be regularly updated, allowing subsequent efforts to map the changing risk scenarios and update or reassess potential damage. The risk scenarios provide useful information on identifying the vulnerable areas and population groups (Sinha, Aditya, & Gupta, 2008).

The main methodological steps in mapping the accessibility in the central area of Bucharest were the following: a) setting up a referenced database of all the buildings with a high seismic risk; b) spatially associating this attribute onto a detailed map

- of the identified buildings; c) determining the indicators regarding the density of buildings, their age, the traffic intensity; d) location of the hospitals, fire stations; e) calculating the accessibility before a possible earthquake; f) identifying potential and specific location of congestion resulting from building; f) determining, by simulation, the inaccessible or poorly accessible areas in a very short time by the intervention crews in case of an earthquake occurrence, taking into account the buildings that might collapse if an earthquake occurs.
- To better know the risks related to earthquakes (which would most likely emanate from the Vrancea area), several types of maps were developed that took into account the region's particular seismogenic characteristics (Mäntyniemi et al., 2003). This approach, focusing on the geophysical phenomena, cannot be applied to achieve the goal that we have set. With this study, GIS is used solely as a tool to determine the accessibility as a starting point for disaster management (Nushi & van Loenen,

2013). Therefore, GIS solutions are demonstrably important applications in understanding the first two phases (risk mitigation and disaster preparedness) of Alexander's (2002) four-phase sequence of emergency management activities.

## 4. Results and Discussion

It is necessary to simulate the emergency interventions prior to the occurrence of catastrophic events because the field situation highlights the importance of factors that need to be kept in mind: the inherited intra-urban structure, with trails of winding streets, their reduced size dating back to medieval times, the poor condition of the buildings, the limited access to important points from where emergency response activities are initiated.

In such a context, the accessibility to a disaster site is extremely important and this requires that such urban areas be treated with special attention. In designing the localization of intervention bases, there should also be taken into consideration the

10 proximity to intensely circulated historic centers. Yet, the biggest impact may be caused by traffic congestion compounded by debris in cases of extreme events, such as earthquakes, which can isolate the critical areas, making impossible a rapid intervention to put down the fires and save human lives.

Inventorying buildings with a high seismic risk (there is a public list of these buildings) combined with precise mapping of their location (Figure 3) and reveals a clear view of their intensity in the historic areas (Figure 4). The most important area is

- 15 the one delimited between Armenească, Moşilor, Călăraşi streets, Splaiul Independenței, Calea Victoriei, Carol and Regina Maria boulevards. The density of these buildings exceeds 2.5 units / hectare and in some areas even 10 units / hectare. In the areas of the highest density, most of the buildings have two or three floors, and because of their uncertain legal status after 1990, they experienced an increasingly advanced degree of degradation. The urban administration and several private entrepreneurs only consolidate maximum 2 buildings per year.
- The density of buildings with high seismic risk shows a very high concentration in the historical center of Bucharest (Figure 5). Looking at a map of the seismicity at the level of Bucharest, it becomes obvious that the inherent risks from earthquake damage are greatest in central Bucharest, including the historical center (Rufat, 2011). Even as most of the buildings located in the historical center date from the early 20th century, they were built on the old foundations of the 19th century (Armaş, 2008).
- To highlight the anticipated degree of access for fire protection and ambulance services in the central area accessibility prior to an earthquake was calculated (and is later compared to a post-earthquake scenario) (Figures 6-7). It is noted that both the firefighter and the ambulance service accessibility is high or very high for the most parts of the capital city, including the downtown area which is especially well served by firefighter and ambulance services. There are 13 large fire stations in Bucharest which can intervene in case of fire (Figure 6).
- 30 Should an earthquake occur, an important consideration is the danger presented by building collapses in obstructing road access. Assessing in detail the high seismic risk buildings led to highlighting, at least for the historic center, the possibility of concentrated building collapses in certain locations (Figure 8). These are the locations that would lead to the isolation of certain buildings and the impossibility of rapid intervention of firefighting or ambulance services.

Thus, the accessibility factor was calculated for the study area and its importance was highlighted to show changes that occur following the increased risk of collapse of the old building stock.

An initial analysis of Figure 9 indicates that the central area seems to be favored due to the possibility of intervention from several points of the city. Superficial observation reveals that most of the fire stations are located around the central area.

- However, this is misleading as pedestrian and vehicular congestion will likely inhibit rapid access of firefighters in some of 5 the areas within the central district. Also, it appears that some areas in the downtown, which previously appeared to have high accessibility for firefighters, are more likely to be impacted by a number of collapsed buildings, potentially blocking the firemen's access (Figure 9), despite the high level of accessibility that was initially perceived.
- Should an earthquake with a magnitude of over 7 degrees on the Richter scale occur, fires can be one of the associated risks. Given that many of these buildings have roofs made of wood and that wood bears an important share of usage as building 10 material, fires are very likely. If action is not taken quickly, in such situations even the likelihood survivors is reduced. In addition, water supply and sewerage systems may be damaged, resulting in basement and ground floor flooding. Firefighting activities must be especially responsive and extremely effective. It would be advisable to provide supplementary emergency response materials at a wider number of locations within this district. This would allow access to such equipment at the local
- scale and as an alternative to other emergency materials and services which may be blocked in the event of an emergency. 15 Assuming that, in the event of a large-scale disaster, certain clusters of buildings may become isolated and inaccessible to emergency services, it is recommended that smaller scale aid stations be established within the district. These smaller scale aid stations could then provide critical assistance in areas blocked by building collapse.
- An in-depth and more detailed of a portion within the general study area shows that while accessibility may appear to be 20 high due to the proximity of a fire station, it may be greatly compromised in the event of multiple cases of building collapse in close proximity to one another (Figure 9). Its position, close to the area of the institutions of national interest (the Parliament building, the buildings of several ministries and other public institutions) leads us to the idea that this is oriented rather on ensuring the protection in case of emergency of these public institutions, than to provide services to an area with such a high density of buildings with high seismic risk.
- 25 Looking at the map which displays both distribution of the high seismic risk buildings and the location of the nearest fire stations (Figure 9), it is clear that there is a need for greater proximity (and hence access) between fire stations and the three areas of maximum density of high risk buildings: one in Lipscani area, another in Bărăției area and the third one in Dorobanți area. The Western area (Grivita - Gara de Nord) could be under the authority of the two existing fire stations. These points of high proximity should be connected to a permanent water supply (as the centralized water sources would have problems due to the earthquake) and also possess a minimum, yet sufficient level of equipment for first response.
- 30

Another important element is accessibility to hospitals and related medical care facilities. Their territorial pattern of distribution at first glance appears to be favorable (Figure 10), but their capacity should be assessed given that the number of casualties could reach a number as high as approximately 11,000. The earthquake of 1940 registered 1271 injured and the 1977 earthquake generated 11,321 casualties (Pavel & Vacareanu, 2015).

To increase the efficiency of emergency response, the assigned areas must be more concretely determined. Otherwise, it can cause an overload for certain hospitals and hence new problems. Additionally, we must consider the availability of specific medical services at individual hospitals and other medical service facilities. The provision of surgical wards, imaging laboratories, and orthopedic facilities is more variable than the the provision of hospitals generally. Depending on the territorial distribution of these hospital services, the buildings with high seismic risk should be assigned so that the accessibility is maximized. Obviously, we discuss dynamic territorial structures, which, depending on the gravity of the reported cases, would include access to other hospitals near the analyzed area.

The population's accessibility to healthcare services is extremely important, even more so in case an earthquake occurs, when the traffic in the affected area increases and its fluidity must be ensured (Toma-Danila, 2013).

10

5

In the event of a powerful earthquake, the communication systems would likely break down either partially or in full, at which point majority of the population tries to get in touch directly with their acquaintances, moving around by car. The quick intervention of the traffic police is absolutely necessary to decrease congestion trends precisely in the areas with the greatest needs of intervention.

- The unpredictable nature of such a phenomenon may lead to bottlenecked traffic at unanticipated locations along the transportation network, which in turn complicates rescue, relief, and evacuation efforts. Under such circumstances, the communication systems between those who are mapping the collapsed or damaged buildings and those who are ensuring the traffic flow must perfectly function and in such a way to allow the wounded to be transported to hospitals and the firefighters' vehicles move towards the critical spots in the city. However, as it is shown in Figure 10, the simultaneous collapse of buildings at the ends of some streets or in smaller intersections (especially in the medieval area of the city) makes it impossible for rapid
- intervention inside the isolated areas. For such situations, the lifesaving equipment, individually transported by persons purposefully trained so, can provide immediate assistance to unlock the main arterial roads.

It is urgently required, however, to renovate the buildings, which present different degrees of degradation and a first necessary step is to analyze and assess the structural characteristics to highlight which buildings that require more immediate attention and to consider the range of intervention measures needed to bring the greatest number of buildings to safety standards

(Tuns, Tamas & Pascan, 2013).

#### 5. Conclusions

GIS can effectively be used as an analytical and decision making tool in planning for hazard mitigation. GIS properly employed provides knowledge concerning emergency response accessibility in areas where physical structures are degraded and pose higher risk of collapse. Such knowledge is critical in anticipating the impact of a disaster. Injury, loss of life, and damage to property is minimized through more effective and rapid emergency response.

Bucharest ranks only second, after Istanbul, in exposure and risk related to earthquakes. It is not enough to be familiar with the distribution mode of the high seismic risk buildings, rather, interventions are critically needed so as to diminish the consequences of such an event. In this respect, the accessibility of rapid intervention means in case of earthquake, in order to

save lives, it becomes a priority. Yet, given the current status and priorities of natural hazard and emergency response planning in the city of Bucharest (and at the national scale, as well), it is unlikely to mitigate the effects of a potential disaster. Across several measures – training of specialists, public awareness and education, infrastructural improvements, and building improvements – efforts are inadequate.

- 5 Perhaps most surprising is the passivity of the urban decision-makers in relation to the very large number of buildings in high risk class. These are concentrated exactly in the most populated and attractive areas in terms of leisure and entertainment. If Bucharest's inhabitants are partially aware of this risk, the tourists, in their vast majority, don't realize what could happen should an earthquake occur.
- The last major disaster event that took place in Romania, the fire in Collectiv club (which led to the deaths of 63 people and seriously injuring around 150 persons), brought to the attention of the local and national authorities the major risk that an earthquake can have. Thus, they banned all shops, restaurants and clubs working in buildings with high seismic risk, but there are not enough resources to rebuild the buildings that continue to be inhabited! The event mentioned above reminded the population and the authorities that a earthquake event or disaster of similar scale will occur at some point and it is necessary to have a clearly defined policy that relies upon concrete measures to reduce the human and material losses.
- Our study reveals the importance of accessibility to buildings for the means of intervention, but also the shortcomings in the provision way of selected emergency response services. These lessons are applicable across numerous cities with similar built environments.

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
