# Peer review of "Mapping Accessibility in the Historic Urban Center of Bucharest for Earthquake Hazard Response"

_Natural Hazards and Earth System Sciences, 2017_

## Referee Comment (RC1) · Anonymous Referee #1 · 8 Jun 2017

**Methodology**

The paper deals with the case study of Bucharest, which is a city vulnerable to earthquakes like a few others which present the Mexico-city effect. The data sets are as well the buildings which are subject of loss, but also items for emergency planning such as hospitals and fire stations. The mapping of the building stock is essential since those with highest risk are likely to collapse and need intervention. The focus lays however on the transportation networks which are subject of the research question on accessibility. To map them, different maps were used for digitalisation in a GIS software. The GIS capabilities permit computing the speed of intervention between the emergency planning items and the collapsed building according to the road and other traffic on the ways. The methodology can be used also for other earthquake prone cities with

similar problems, taking into account particularities such as the fact that Bucharest is located on a plain. The presence of both historic city centre with narrow streets and later boulevard development from the 19th century enhances the validity. The variables are well defined and measured with means designed for this in a GIS environment for road network analysis. There are other studies from earlier years when such methods had to be first programmed in a software (Fiedrich, Goretti). A review of these and also a connection to Space Syntax remains unclear. Also the relationship to agent based modelling could be more clear.

**Data and Results**

The study matches the results as presented briefly in the abstract. The paper properly underlines the study results with a table and graphs. The graphs present accessibility before and after an earthquake, for both fire fighters and ambulances. The reduced accesibility is calculated through the blocking of roads through collapse of high seismic vulnerability buildings, which are at the same time for Vrancea earthquake high rise buildings. Fire after earthquake is considered a major threat as the roofs are out of wood, but this is not the case for these high vulnerability buildings. This should be corrected. The accessibility from more emergency nodes is also discussed. The text presents the discussion of the data in the figures, not their description. Given that the area chosen for the study is relevant through the density of high seismic risk buildings, this is a statistically relevant result.

---

## Author Comment (AC1) · 28 Jun 2017

The authors highly appreciate the interactive comment related to the approach of the measuring accessibility in a congested urban area that could be affected by an earthquake. Thanking to anonymous referee, the authors consider the comment very interesting and this was deeply analyzed. Our direct answers are the followings:

a) In relation with the comment that "fire after earthquake is considered a major threat as the roofs are out of wood", we know very well that excepting the "Hanul lui Manuc" complex, the majority of buildings have sheltered covers. But the roof board is fixed to a wooden stand, so the wood is dominant on the roof. Analysing the referee'suggestion we will improve the phrases, underlying that the majority of buildings have wood struc-

ture and/or components (some buildings from "Selari" – see "Crama Domneasca", "Covaci" and "Smardan" streets, for example). At the same time, we will add the fact that each building has restaurants, cafes or pubs, which means a huge quantity of furniture, an important source for fire.

b) The authors will develop, in the final form, the description of the data about high seismic risk buildings; this suggestion could increase, indeed, the relevance of the study.

c) Regarding the statistical data approach to obtain relevant results, it should be mentioned that the study is focused on highlighting the degree of accessibility as a pattern in two distinct situations: before and, especially, after the earthquake. For the next topic development, we intend to go in-depth study, collecting more statistical information and to make the step to scenarios' elaboration (for example, blocking the crossroads X or Y, by the collapse of building A or B, what is happens?). The idea of the referee is very good one and we will take into consideration in our future research.

d) Thank you so much to recommend us the Goretti (including his collaborators) works! Indeed, he has interesting approaches of earthquake management (adapted to the specific of Italian earthquakes, especially), and we will cite him in the final form. For example, some ideas from his paper "The Urban System of Crotone, Italy, Facing the Earthquake Impact" published in Bostenaru Dan, Armas and Goretti: Earthquake Hazard Impact and Urban Planning, Springer (2014) we find very useful. It's true that there are many other authors, who have had an important contribution to a better knowledge of the earthquake mechanisms and management, but the limited space and the aim of our paper have reduced the development possibilities.

---

## Referee Comment (RC2) · Anonymous Referee #2 · 24 Jul 2017

**Referee report on ms nhess-2017-13 "Mapping Accessibility in the Historic Urban Center of Bucharest for Earthquake Hazard Response", submitted to Natural Hazards and Earth System Sciences Discussion**

July 24, 2017

The authors present a study on post-disaster accessibility of the historical centre of Bucharest using an earthquake scenario affecting the city. Accessibility was computed by means of GIS using the geometric structure of the transport network, and taking into account possible network interruptions as a result of a seismic event. The accessibility was calculated as a function of the distances between different residential areas and hospitals areas and of the time necessary for these movements using isochrones. As such, the topic is of relevance for the target journal. However, before the manuscript may become acceptable for publication in NHESS, the following shortcomings should be addressed in order to increase the accessibility of the work.

**0.0.1 *Introduction**

The introduction of the article outlines the importance of risk management for seismic hazards, the preparation of precise emergency plans and the use of GIS methods to obtain these plans.

- Please carefully review the consistent use of definitions and formulation in this section (e.g. p.2. l.3-5, p.2 l.13-14 or p.2 l.27-28).

**0.0.2 *Case Study**

The case study fits the topic in general because of the described high exposure for earthquakes and the dense urban structure in the city core of Bucharest with an old building stock. The description of the case study is detailed.

- The text and the caption of Fig.2 refers to a different year (1972 and 1977)

- Statements such as p.3 l.15-16 need an explanation: Why is it interesting for this study?

**0.0.3 *Data and methods**

The section presents the used data sets and methods of the study. I think some points of these paragraphs need more explanation to make the study fully understandable.

- The authors may wish not to over-emphasise the use of GIS in their analysis since this is a tool widely accepted in the research community (and in practice) for geospatial analysis.

- The description of the used data is insufficient. Please name the sources of the datasets and their background. Especially a description of the dataset "buildings subject to seismic risk" is missing. How seismic risk for this buildings was calculated? Is it only a binary dataset? Was seismic risk analysed using an appropriate earthquake model?

- The methods section (p.5) is not really coherent. Please assign the different tools to the described working steps (p.5 l.23-29) to make your research reproducible.

- Ad. step f): How did you identify locations of congestions? Please describe the workflow.

- Ad. step g) (in the text f) again): Do you follow the assumption that every building from your dataset will collapse in case of an earthquake? So is it a one scenario/worst-case analysis? Can you establish a connection between the intensity of an event and the collapsing buildings (based on structure, age, exposure,...)?

- It remains unclear how the described traffic data was added in the analysis?

0.0.4 *Results and Discussion*

The section presents the results of the network analysis. The accessibilities before and after an earthquake event are briefly discussed.

- Fig.3, Fig.4 and Fig.5 all show the distribution of the building stock with a high seismic risk. Fig.3 is very hard to read and is not really necessary when there is Fig.5. Fig.4 additionally shows the traffic. A big improvement would be the homogenization of Fig.4 and Fig.5 regarding the scale and the extent of the maps to make them comparable. Please reconsider the colour ramp of Fig.5 – the map has too many classes and no intuitive colouring.

- Please name what density estimation you used for Fig.5 in the text or figure caption

- Fig.6 and Fig.7: Why you use a 3D visualisation? Please do not use a scale bar in a 3D map. To improve the figures also adapt scale and extent to the same as in Fig.5 and make the maps simply 2D.

- For Fig.6 and Fig.7 there is no description how you calculated the results, is it a casual service area analysis?

- The references to the Figures do not fit on p.7, there is no Fig.10.

- The results of Fig.8 and Fig.9 are comprehensible. How you got these results, please add a description of the approach in the text – otherwise I have to emphasise that the methods are not fully mirrored in the results and as such the paper cannot be accepted. Why there are areas in the direct neighbourhood of hospitals or firefighters with a low accessibility, although there are no buildings at risk (around Kogalniceanu Square)? And areas far away from the station with a lot of buildings at risk around that area with a high accessibility (north and east of Romana Square)?

- Improve Fig.8 and Fig.9 by homogenisation of scale, extent and look.

- Please indicate given region names (p.7 l.27) on one of the maps.

0.0.5  *General notes*

- The application of network analysis for mapping accessibilities is common sense. The methodological approach of the study utilizes buildings with seismic risks as potential road blocks in case of an earthquake event. To improve the method the application of different scenarios (based on earthquake intensity or building indicators) is recommended.

- The part of the study with policy recommendations is rather short, how the produced maps and results can be used in the future?

- The article does not discuss validity and accuracy.

- Moreover, I kindly would like to recommend a sound proofreading of the manuscript by a native speaker, even if proofing is also offered by Copernicus.

These shortcomings should be carefully addressed before the material may become acceptable for final publication in NHESS. Therefore, I recommend major revisions.

Interactive
comment